# Challenges in Total Hip Arthroplasty with Prior Spinal Arthrodesis: A Comprehensive Review of Biomechanics, Complications, and Surgical Strategies

**DOI:** 10.3390/jcm13113156

**Published:** 2024-05-28

**Authors:** Riccardo Giai Via, Francesco Onorato, Michele Reboli, Stefano Artiaco, Matteo Giachino, Francesco Bosco, Alessandro Massè

**Affiliations:** 1Department of Orthopaedic and Traumatology, Orthopaedic and Trauma Center, University of Turin, 10125 Turin, Italy; riccardo.giaivia@unito.it (R.G.V.); francesco.onorato@unito.it (F.O.); michele.reboli@unito.it (M.R.); stefano.artiaco@unito.it (S.A.); matteo.giachino@unito.it (M.G.); alessandro.masse@unito.it (A.M.); 2Department of Precision Medicine in Medical, Surgical and Critical Care (Me.Pre.C.C.), University of Palermo, 90133 Palermo, Italy; 3Department of Orthopaedics and Traumatology, G.F. Ingrassia Hospital Unit, ASP 6, 90131 Palermo, Italy; 4Department of Orthopedic and Traumatology (DICHIRONS), University of Palermo, 90127 Palermo, Italy

**Keywords:** total hip arthroplasty, THA, spinal arthrodesis, biomechanics, complications, surgical strategies

## Abstract

Total hip arthroplasty (THA) has revolutionized patients’ lives with hip osteoarthritis. However, the increasing prevalence of THA in individuals with prior lumbar arthrodesis (LA) poses unique challenges. This review delves into the biomechanical alterations, complications, and surgical strategies specific to this patient subset, highlighting the need for tailored preoperative assessments and planning. Due to altered pelvic and spinal biomechanics, patients with LA undergoing THA face a higher risk of dislocation and revision. The complex interplay between spinal and hip biomechanics underscores the need for meticulous preoperative planning. Comprehensive clinical examination and radiographic evaluation are vital for understanding patient-specific challenges. Various radiographic techniques, including computed tomography (CT)/X-ray matching and standing/seated studies, provide insights into postural changes affecting pelvic and spinal alignment. Complications following THA in patients with LA highlight the necessity for personalized surgical strategies. Careful consideration of implant selection, the surgical approach, and component positioning are essential to prevent complications. In summary, THA in patients with prior LA demands individualized preoperative assessments and planning. This approach is crucial to optimize outcomes and mitigate the heightened risks of complications, underlining the importance of tailored surgical strategies.

## 1. Introduction

Total hip arthroplasty (THA) is a highly effective surgical procedure for osteoarthritis patients with chronic hip pain or other pathologies requiring joint replacement [1,2]. In most cases, THA allows early pain relief, improved functional outcomes, and overall enhancement of the quality of life [1,2,3,4]. It is not so rare for patients undergoing THA surgery to have previously undergone lumbar arthrodesis (LA) [4,5,6]. This condition has seen a notable increase, likely attributed to the widespread adoption of spinal stabilization, THA techniques, and demographic shifts such as an aging population [7,8]. The recent literature has reported a 293% rise in patients undergoing THA and LA surgery over the past decade compared to patients treated only with a THA without LA [1,2,3,6,7,8].

Patients with prior spinal stabilization have a higher risk of dislocation and revision after THA surgery compared to those without prior LA [2,3,6,7,8,9,10]. LA, the fusion of the lumbar spine, may modify the biomechanics of the pelvis and spine, especially during postural changes [11,12]. Compensatory mechanisms may increase femoral mobility in response to these biomechanical alterations. However, this may cause an accompanying risk of anterior and posterior impingement, significantly raising the susceptibility to dislocation [6,7,8,13]. Additionally, the altered biomechanics can lead to increased wear and tear on the prosthetic components, potentially resulting in earlier failure and the need for revision surgery.

Given these complexities, managing patients with a history of LA undergoing THA requires a multifaceted approach. This involves not only addressing the immediate surgical concerns but also considering long-term outcomes and patient-specific factors that can influence recovery and functionality. Advanced imaging techniques, personalized surgical planning, and innovative implant designs are crucial in optimizing results for these patients.

This review aims to explore the distinctive challenges presented by THA in the context of prior LA. Emphasizing the indispensability of clinical examination, comprehensive review of imaging studies, and meticulous preoperative planning, it endeavors to mitigate the risk of postoperative complications and ensure enduring stability, longevity, and functional efficacy of hip replacement. By integrating recent advancements in surgical techniques and understanding the biomechanical interplay between the spine and pelvis, we aim to provide a thorough overview of strategies to improve patient outcomes.

## 2. Biomechanics

The biomechanics of the pelvis and spine are intricately linked to the extent that Dubousset coined the term’ pelvic vertebrae’ [14]. Key to this understanding are the concepts of pelvic incidence (PI), pelvic tilt (PT), and sacral slope (SS). These relationships form the basis for comprehending spine alignment and musculoskeletal function [13,14,15].

The PI represents the angle formed by a line drawn from the femoral head’s center to the sacral endplate’s midpoint and a line perpendicular to the sacral endplate. The PI, a unique anatomical parameter for each individual, plays a crucial role in determining the overall sagittal balance of the spine. The PI results are based on the sum of the angles SS and PT.

The PT refers to the pelvis’s orientation relative to the femoral heads. It is the angle formed by a line connecting the midpoint of the sacral plateau to the center of the femoral heads and the vertical. PT reflects the pelvis’s forward or backward tilt and influences lumbar lordosis (LL). PT is inversely proportional to the SS.

The SS is the angle between the horizontal and the sacral endplate. It represents the degree of SS and influences LL. Together with the PT, the SS contributes to the overall sagittal alignment of the spine. The SS is inversely proportional to the PT.

SS and PT are not static but dynamic and constantly related. They tend to vary as we transition from standing to sitting. In the standing position, PT decreases with a higher SS, while in the sitting position, PT is greater than the SS (Figure 1). This dynamic relationship is crucial to understanding the biomechanics of the pelvis and spine. 

During the transition from standing to sitting, acetabular anteversion (AA) increases in a constant ratio: each 1° of retroversion (i.e., increase in PT) corresponds to a 0.7° increase in AA [4,5,6] (Figure 2).

Considering the concept of Dubousset of pelvic vertebrae, as the standing and sitting positions vary, the LL also varies [14]; in the transition from standing to sitting position, in addition to the retroversion of the pelvis, there is a concomitant reduction in LL.

PT and PI are interrelated. This means there is a correlation between PI and LL, as explained by Schwab et al. and then by Le Huec et al., referred to as PI-LL mismatch [16,17]. In addition, Phan et al., considering PT and PI-LL mismatch, divided columns into balanced (if PT < 25° and PI-LL < 10°) and unbalanced (if PT > 25° and PI-LL > 10°) considering sagittal alignment and, in turn, are further divided into flexible and rigid [7]. Following this concept, patients with LA may be considered rigid and balanced or unbalanced based on sagittal spine alignment after surgery. More simply, when the lumbar spine is fused by arthrodesis, the ability of the spine to increase and decrease lordosis during postural changes is compromised. This limitation extends to the pelvis, affecting the mechanisms of pelvic change during the postural adjustments mentioned earlier [7,8]. LA significantly affects PT and SS adjustments during postural changes, as it involves the surgical fusion of two or more vertebrae in the lumbar spine, limiting the spine’s flexibility and ability to adapt to different postures.

Usually, the pelvis tilts anteriorly or posteriorly to help maintain balance and the body’s center of gravity during activities such as standing, sitting, and walking. However, with lumbar fusion, the pelvis’s ability to adjust its tilt is compromised, resulting in a fixed pelvic position. This may lead to a condition known as “stuck standing” or “stuck sitting”.

In the case of “stuck standing,” the pelvis is typically locked in an anterior tilt position with reduced PT and SS. This makes it difficult for the pelvis to rotate posteriorly, which is necessary for sitting comfortably. It causes difficulty transitioning from standing to sitting and increases the risk of posterior hip dislocation due to anterior impingement.

Conversely, in the case of “stuck sitting,” the pelvis is locked in a posterior tilt position with increased PT and SS. This may cause problems in achieving an upright standing posture and increase strain on the lower back and hips, resulting in an increased risk of anterior hip dislocation due to posterior impingement.

In essence, the patient’s pelvis may be locked in two ways: as if they are always standing or “stuck standing” thus with reduced PT and reduced AA (Figure 3), or as if they are always sitting or “stuck sitting”, with increased PT and AA (Figure 4) [6].

Suppose a patient presents with a “stuck standing” pelvis. In that case, they will tend to dislocate posteriorly more often in postural changes from sitting to standing due to an anterior impingement, whereas in “stuck sitting” pelvis, the dislocation tends to be more anterior secondary to a posterior impingement [8,11].

## 3. Clinical Examination

A comprehensive approach is essential when examining a patient with a history of LA in preparation for THA surgery. Begin with a careful assessment of the lumbar spine, paying attention to any stiffness or range-of-motion (ROM) limitations related to the arthrodesis and the degree of LL in the sagittal plane. Examine the pelvis for alignment irregularities that could affect the hip replacement and its position, whether anteverted or retroverted in the sagittal plane. Observing the patient’s gait may help identify compensatory movements influenced by lumbar issues [6,9,10,13].

Detect that there are no peripheral vasculo-nervous deficits. Perform the Trendelenburg test and inquire about and assess pain, distinguishing between lumbar-related discomfort and potential hip pain [9,10]. Assess the patient’s overall status and functional demands, considering daily activities and mobility in the context of LA and the potential problems this may bring in a THA surgery [10,13].

Differentiating lumbar from hip-based pain is critical. The following criteria, drawn from guidelines by the American Academy of Orthopedic Surgeons (AAOS) [1,15], may aid in this differentiation (Table 1).

Furthermore, it includes an assessment of lower limb mechanics, evaluating how limitations in the lumbar spine might affect leg function and gait [6,7,9,10,13]. Examine the alignment and ROM of the knees and ankles, as compensatory mechanisms due to restricted lumbar movement may lead to abnormal stress and potential pathology in these joints. Assess muscle strength and flexibility in the lower limbs, particularly the quadriceps, hamstrings, and calf muscles, as imbalances can influence overall mobility and stability [9,10,13].

Evaluate the patient’s walking pattern, looking for signs of limping, uneven stride length, or altered foot placement, which may indicate compensatory strategies or underlying issues [6,7,9,10,13]. Using gait analysis tools may provide a detailed understanding of how lumbar spine limitations impact lower limb function [9,10,13].

Consider how limitations in hip ROM, influenced by LA, may affect the selection of surgical techniques. For example, restricted hip flexion and extension may necessitate modifications in the surgical approach to ensure proper component positioning and avoid impingement [10,12,13]. An anterior approach might be favored to allow better access and visibility of the hip joint, facilitating accurate placement of the acetabular and femoral components. Alternatively, a posterior approach may be chosen to minimize the risk of dislocation in patients with severe lumbar rigidity. Tailoring the surgical technique to the patient’s specific biomechanical limitations is crucial for optimizing surgical outcomes and minimizing postoperative complications [10,12,13].

## 4. Radiological Assessment

As with all patients scheduled for prosthetic hip replacement surgery, radiography is mandatory for diagnostic purposes and effective preoperative planning. However, in patients with LA, taking only an AP radiograph of the pelvis with anteroposterior (AP) and later-lateral X-rays of the hip requiring surgery may not allow proper calculation of PI, PT, SS, and especially LL degrees. There is no standardized radiographic protocol for these patients in the literature, so several solutions have been described. Inoue et al. propose a study using the CT/X-rays matching technique of the pelvis [18]. Other authors, however, such as Goyal et al. and Furuhashi et al., proposed a standing and seated radiographic study of the pelvis and spine together in AP and lateral-lateral X-rays to examine postural changes where there is likely to be the most significant risk of dislocation [11,19]. Behery et al. and Schmidth Braekling et al. propose the use of combined AP and lateral-lateral X-rays of the pelvis and spine in other positions in addition to standing and sitting, such as maximum trunk flexion position while seated or single-leg standing visualization, which emphasizes the compensatory mechanisms of patients and may provide information on patients at high risk of dislocation in THA. 

Some researchers have extended this methodology to include using EOS radiography, a modern imaging modality characterized by low radiation doses. EOS facilitates comprehensive evaluation of the pelvis and spine in both coronal and sagittal planes, accommodating assessments in standing, sitting, and other positions. This holistic approach offers valuable insights into biomechanical adaptations and aids in identifying patients who may require tailored interventions to mitigate the risk of post-THA complications [6,7].

## 5. Influence of Surgical Approach

The choice of the surgical approach is a crucial factor that can significantly impact the outcomes of THA for patients with prior LA [18,19,20]. 

The anterior approach, which involves accessing the hip joint from the front, offers a direct view of the acetabulum and femur, enabling precise component placement. This approach is particularly advantageous in patients with lumbar fusion, as it allows for intraoperative adjustments to accommodate the fixed spinal-pelvic alignment, therefore reducing the risk of impingement and dislocation [18,19].

The posterior approach, accessed through the back of the hip, is a familiar technique among surgeons. It may be advantageous for its reduced risk of anterior impingement. However, careful attention to the positioning of the components is required to avoid dislocation, especially in patients with altered spinal-pelvic dynamics due to lumbar fusion [18,20,21,22]. 

In some cases, the direct lateral approach may be considered. It offers a balance between anterior and posterior techniques, with a reduced risk of dislocation and good access to the hip joint [19].

In patients with significant lumbar spine rigidity, advanced techniques such as computer-assisted navigation or robotic surgery may enhance precision in component placement, ensuring alignment that compensates for the lack of lumbar flexibility [18,19,20,21]. Dual-mobility implants, which provide a more extensive range of motion and increased stability, may also be considered to mitigate the heightened risk of dislocation in this patient population [18,19,23].

By incorporating these additional assessments, clinicians can develop a more comprehensive treatment plan that addresses the lumbar spine and lower limb mechanics, ultimately improving outcomes for patients undergoing THA surgery [9,10,13]. This focused physical examination provides a nuanced understanding of the patient’s condition, contributing to informed decision-making and adequate preparation for THA surgery [6,7,9,10,13].

## 6. Discussion

Many studies analyzed the biomechanical changes and the adaptation mechanisms adopted by our body in patients with LA [7,8,14,15,16,17]. The studies are all aligned in that patients who underwent THA with a previous LA generally demonstrate positive results but with more complications than patients who operated for hip replacement without a previous LA [7,8,14,15,16,17].

Complications of THA surgery may significantly impact the patient’s quality of life and the surgical outcome. Several studies have analyzed the incidence of the most frequent complications following isolated THA surgery or combined with LA, such as hip dislocation, aseptic loosening, superficial wound infections, and PJI [14,15,16,17]. In our recent systematic review and meta-analysis, we showed superior outcomes in patients who underwent THA (number of patients included: 3.081.137) compared to patients who received THA after a previous LA (number of patients included: 58.027) [10]. Specifically, patients with isolated THA implants showed a significantly lower incidence of THA dislocation, wound complications, PJI, revision, mechanical complications, and aseptic loosening. It revealed a significant overall advantage for THA alone.

Similarly, other authors [13] previously conducted a meta-analysis and systematic review, showing that PROMs were lower and the rate of dislocations, revisions, and all other complications was 5.4, 6.3, and 4.6 times higher in the LA group, respectively. 

Anterior or posterior dislocation is one of the most common complications and may lead to an unstable implant, which indicates revision [22,23].

One reason for the higher rate of complications in patients undergoing LA before THA surgery is likely the biomechanical changes caused by vertebral arthrodesis, which explicitly result in a reduced PT and AA.

Although several factors may promote THA dislocation, including osteophytes, excessive scar tissue, and suboptimal positioning of prosthetic elements, the cup and stem positioning is crucial in preventing hip dislocation. Historically, Lewinnek et al. [24] asserted that the “safe zone” for THA consists of 15° ± 10° of AA and 40° ± 10° of cup inclination. The transverse acetabular ligament is a reference during surgery to guide the cup implantation within Lewinnek’s plane. However, in the presence of altered PT, relying solely on the transverse acetabular ligament becomes less reliable, as each degree of posterior PT corresponds to 0.7° of cup anteversion, resulting in inaccurate cup placement [25,26]. Because there is a loss of physiological pelvic adaptation mechanisms in the patient undergoing LA in standing and sitting positions, cup, and stem positioning might be even more crucial [27,28] and might be customized to specific patient anatomy and biomechanics to prevent complications. 

The complex interplay between spinal and hip biomechanics and the demonstrated superior incidence of THA dislocation in patients with prior LA necessitates detailed preoperative planning and a personalized, patient-specific approach [25,26,27,28]. Recent literature on surgical strategies fails to demonstrate any difference between the anterior and posterior approaches in terms of change in postoperative PT, as shown by Inoue et al. [18]. However, Kahn et al., in a comprehensive retrospective analysis within their institution, showed that in patients with lumbar spondylosis or lumbar fusions, utilization of an anterior approach was associated with a lower risk of subsequent dislocation [29]. Additionally, both the direct anterior and direct lateral supine-based approaches showed low dislocation risk without any difference compared to patients without LA undergoing THA [19,30]. The supine patient position enables intraoperatively fluoroscopic checks of cup component version and inclination, facilitating a more accurate component positioning to mitigate uncertainties.

The spinopelvic and hip joints act like two complementary hinges: the counterpart compensates for any reduction or augmentation of the proper ROM [31]. Therefore, acetabular cup implants in patients with previous LA should be carefully assessed and planned, as they are crucial in preventing hip dislocations. The historical “Lewinnek safe zones” are often insufficient, and positioning should be tailored according to the patient’s sagittal posture and biomechanics. Patients affected by the “stuck sitting” phenomenon have a persistently posteriorly tilted pelvis. Therefore, while standing, the femur must hyperextend for balance, and the combination of motions results in an increased risk of posterior impingement and anterior dislocation due to the alteration of the combined sagittal indexes. In this biomechanical pattern, acetabular components should be positioned with an inclination of 35–40° and anteversion of 15–20°, preferably with an increased offset of 5 mm to prevent posterior trochanter impingement [32]. 

Conversely, patients with the “stuck standing” phenotype have an anteriorly tilted pelvis. Therefore, the femur must flex more when sitting, posing a risk of bony impingement of the anterior trochanter and subsequent posterior dislocation, especially during postural changes. The inclination of 45–50° and anteversion of 20–25° should be considered to reduce the risk of posterior dislocation for these patients.

During the decision-making process, a personalized, patient-specific approach that evaluates the patient’s age and functional activities must be performed, as an inclination over 45° could be associated with increased insert wear [33].

Among the surgical options to prevent THA dislocation, large head diameters of the femoral component have demonstrated a reduced dislocation risk, even in THA with LA [34,35].

Larger femoral head diameters allow for a greater ROM as the hip flexion angle required for dislocation is directly proportional to the diameter of the femoral head; a diameter larger than 28 mm is associated with a lower risk of dislocation [34,35].

Additionally, dual-mobility (DM) constructs providing a greater ROM to impingement and a more significant jump distance have become attractive in primary and revision THA patients at high risk of impingement and dislocations [36]. In Chalmers et al.’s study, no patients treated with THA after LA experienced any dislocations at a mean of 3 years of follow-up, despite the variability in the acetabular component position [37]. Therefore, DM constructs could be considered an attractive option for these patients to decrease dislocation rate and increase impingement-free stability in high-risk LA patients.

## 7. Strengths and Limitations

The present study provides a comprehensive overview of the increasing trend of THA combined with LA; it highlights that patients with prior LA are considered a high-risk group for complications during THA surgery, particularly dislocation and subsequent revision. The review also gives attention to the discussion on pelvic and spinal biomechanics to better understand the impact of LA on PT, AA, and LL, aiding recognition of the complexities of THA in these patients. Overall, this study also sheds light on surgical strategies like anterior and posterior approaches, large femoral head diameters, or dual-mobility constructs to mitigate complications in THA with prior LA.

Although this review is intended to offer a comprehensive analysis of the available literature, it is important to note several limitations. First, since this study is a narrative review, the selection of studies may have introduced biases due to the limited availability of research articles on some aspects of the topic. Despite efforts to include all relevant studies, the possibility of overlooking important ones due to search criteria or other limitations cannot be ruled out. Second, the quality of the included studies varies, potentially affecting the overall conclusions drawn from this review. Caution should be exercised when interpreting the results. Third, the scope of this review is limited to studies published up to the current date, so new research may emerge that challenges or extends the results presented here. Lastly, it is essential to recognize that this review summarizes the existing literature and does not involve collecting or analyzing original data.

## 8. Conclusions

Patients who have undergone LA and are now going for THA should receive more comprehensive treatment. The preoperative study should include a detailed clinical and radiographic examination of the pelvis and lumbar spine in different positions. By conducting a comprehensive radiographic survey, it is possible to identify the areas where the patient is more prone to dislocate. This allows for more precise preoperative planning, enabling the surgeon to decide on the appropriate access to be used, the type of implant required (such as a prosthesis, double mobility, or large femoral heads), the degree of tilt and anteversion to be given to the cup. This is particularly important in patients who have undergone LA in the past, as they have higher rates of complications. In addition, in the future, the use of navigation systems and robotics in THA for patients with previous LA could lead to greater accuracy in optimal implant placement and, thus, better outcomes with lower complication rates.

## Figures and Tables

**Figure 1 jcm-13-03156-f001:**
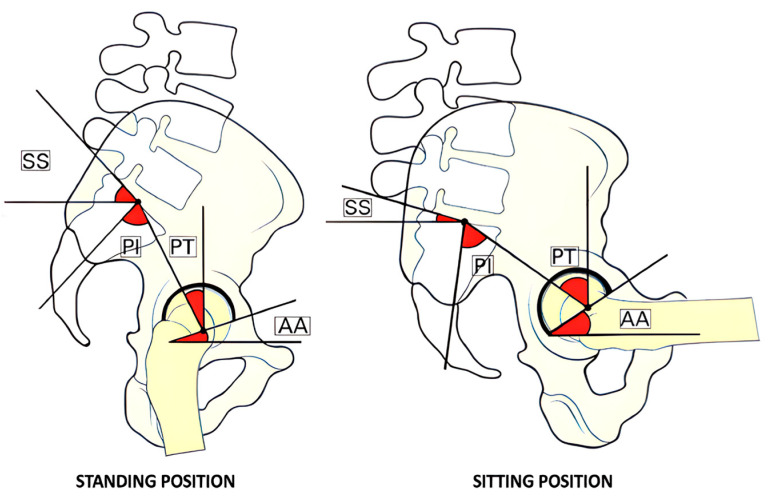
This image shows the variability of angles between the usual positions of the pelvis in standing and sitting. The representation on the left shows a conventional standing posture characterized by an anteverted pelvis with low pelvic tilt (PT), high sacral slope (SS), and minimal acetabular anteversion (AA). In contrast, the image on the right illustrates a retroverted pelvis with high PT, high AA, and low SS. The abbreviations are PT for pelvic tilt, SS for sacral slope, AA for acetabular anteversion, and PI for pelvic incidence.

**Figure 2 jcm-13-03156-f002:**
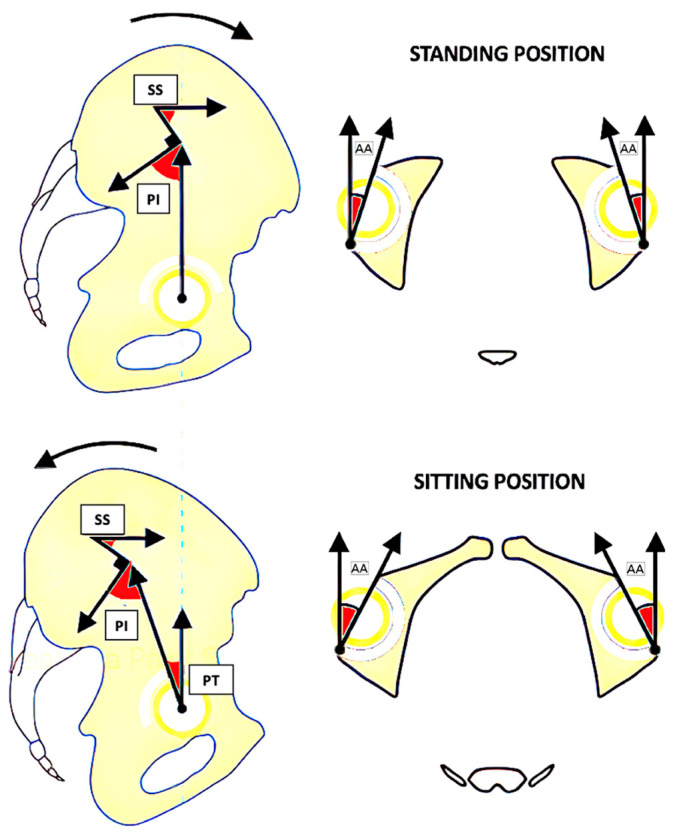
This image shows the correlation between pelvic tilt (PT) and acetabular anteversion (AA). The upper two images show how the pelvis is positioned while standing and how AA is reduced. In the sitting position, on the other hand, as shown in the bottom two figures, the pelvis is retroverted, and the degree of AA increases. The abbreviations are PT for pelvic tilt, SS for sacral slope, AA for acetabular anteversion, and PI for pelvic incidence.

**Figure 3 jcm-13-03156-f003:**
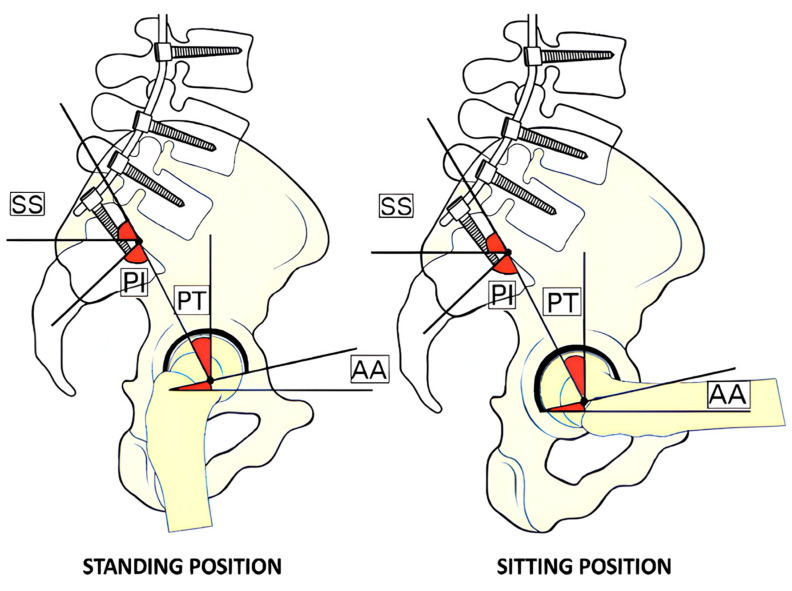
Stuck standing. This illustration shows the impact of spinal arthrodesis on typical biomechanics in stuck standing. Note that changes in standard angles during postural adjustments are minimal. The pelvis maintains a stabilized position, with constant pelvic tilt (PT), sacral slope (SS), and acetabular anteversion (AA). The abbreviations are PT for pelvic tilt, SS for sacral slope, AA for acetabular anteversion, and PI for pelvic incidence.

**Figure 4 jcm-13-03156-f004:**
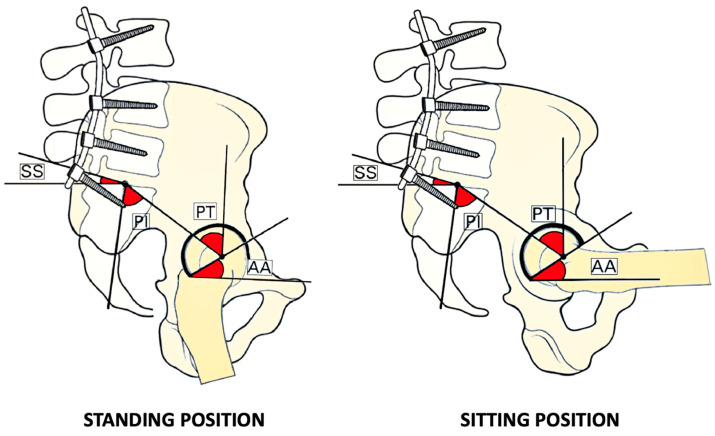
Stuck sitting. This illustration shows the impact of spinal arthrodesis on typical biomechanics in stuck sitting. Note that changes in standard angles during postural adjustments are minimal. The pelvis maintains a stabilized position, with constant pelvic tilt (PT), sacral slope (SS), and acetabular anteversion (AA). The pelvis is in both conditions retroverted, and the lumbar lordosis is reduced. The abbreviations are PT for pelvic tilt, SS for sacral slope, AA for acetabular anteversion, and PI for pelvic incidence.

**Table 1 jcm-13-03156-t001:** Criteria for Differentiating Lumbar Pain from Hip Pain.

AAOS Criteria	Lumbar Pain	Hip Pain
Location of Pain	Lower back, may radiate to buttocks, thighs, and feet	Groin, thigh, buttock, may extend to the knee but rarely beyond
Nature of Pain	Dull ache or sharp, shooting pain, particularly if nerve involvement (sciatica)	Deep, aching pain, exacerbated by weight-bearing activities
Pain Triggers	Worsens with prolonged sitting, standing, or bending forward; improves with walking or reclining	Worsens with walking, climbing stairs, or hip rotation
Range of Motion	Limited lumbar spine movement, particularly during flexion or extension	Restricted hip ROM, especially in internal and external rotation
Physical Examination	Reproduced with specific maneuvers like the SLR test or by palpating the lower back	Elicited by direct palpation of the hip joint or tests like the FABER test

AAOS: American Academy of Orthopedic Surgeons; ROM: Range of motion; FABER: Flexion, Abduction, and External Rotation; SLR: straight leg raise.

## Data Availability

The data presented in this study are available in the article.

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
