# Peer review of "Challenges in Total Hip Arthroplasty with Prior Spinal Arthrodesis: A Comprehensive Review of Biomechanics, Complications, and Surgical Strategies"

_jcm, 2024, doi:10.3390/jcm13113156_

Round 1
Reviewer 1 Report
Comments and Suggestions for Authors
Please concisely use abbreviations. There is no point in using both THA and THR in the same text. All abbreviations should be defined or spelled out at first appearance in text.
Figure 3 should clearly state that it refers to a “stuck standing” position. Maybe a set of illustrations showing the “stuck sitting” issues would add to the paper's clearness.
Lines 123-126: Is there any evidence on this supposition?
Clinical examination paragraph: The suggestions mentioned here are often propaedeutic or standard protocol for hip examination or history taking. Please limit to aspects that offer a unique approach to the problems that may arise in the context of lumbar arthrodesis.
Discussion paragraph: The name is not “Lawinnek” it is Lewinnek as correctly mentioned in reference 24.
Comments on the Quality of English LanguageIn general, the text is clear and informative, but there are a few areas where clarity, conciseness, grammar, and flow can be improved. Please consider professional native speaker input.
As example, lines 170-172, “tendentially” is not a standard word in English. Also, “who operated for” should be “who underwent.”
Author Response
Thank you for your email with the reviewers’ comments. We appreciate the helpful feedback from the reviewers on improving the quality and content of this manuscript. The manuscript has been revised accordingly. We have addressed the specific reviewer concerns in a point-by-point manner below. We hope that this revised manuscript is now suitable for publication and look forward to hearing from you.
Best Regards

Reviewer 2 Report
Comments and Suggestions for Authors
- "Total hip arthroplasty (THA) is a highly productive intervention" - "productive" is unscientific in this context
- the title is comprehensive
- "It is not so rare" - avoid this type of expressions
- The 293% increase is presented without a comparison. Authors should provide baseline figures or clarify what this percentage increase is compared to
- there are many situations in which the authors introduce some abbreviations multiple times in the manuscript
- line 107 onwards - the authors should explain how lumbar fusion affects pelvic tilt and sacral slope adjustments during postural changes
- the authors should also detail on the examination - include an assessment of lower limb mechanics and how lumbar spine limitations might affect leg function and gait
- row 135 - how limitations in hip ROM, influenced by lumbar arthrodesis, can affect the selection of surgical techniques?
- Include a brief guide or criteria on differentiating lumbar from hip pain based on patient history - maybe some guidelines from the literature or from specific societies
- references are up to date
- figures are clear and concise, with proper legends
Author Response
Thank you for your email with the reviewers’ comments. We appreciate the helpful feedback from the reviewers on improving the quality and content of this manuscript. The manuscript has been revised accordingly. We have addressed the specific reviewer concerns in a point-by-point manner below. We hope that this revised manuscript is now suitable for publication and look forward to hearing from you.

Reviewer 3 Report
Comments and Suggestions for Authors
This is a well written paper which contains a narrative review about pelvic-hip relationship after lumbar arthrodesis. The work is weel organized with a good discussion.
I suggest to add a specific paragraph to investigate how surgical approach can influence this issue (several studies has been recently published about this topic).
Author Response

(The authors gave the same response as above.)
